# The Association between Peptic Ulcer Disease and Gastric Cancer: Results from the Stomach Cancer Pooling (StoP) Project Consortium

**DOI:** 10.3390/cancers14194905

**Published:** 2022-10-07

**Authors:** Pedram Paragomi, Bashir Dabo, Claudio Pelucchi, Rossella Bonzi, Abdulaziz T. Bako, Nabila Muhammad Sanusi, Quan H. Nguyen, Zuo-Feng Zhang, Domenico Palli, Monica Ferraroni, Khanh Truong Vu, Guo-Pei Yu, Federica Turati, David Zaridze, Dmitry Maximovitch, Jinfu Hu, Lina Mu, Stefania Boccia, Roberta Pastorino, Shoichiro Tsugane, Akihisa Hidaka, Robert C. Kurtz, Areti Lagiou, Pagona Lagiou, M. Constanza Camargo, Maria Paula Curado, Nuno Lunet, Jesus Vioque, Paolo Boffetta, Eva Negri, Carlo La Vecchia, Hung N. Luu

**Affiliations:** 1Division of Cancer Control and Population Sciences, UPMC Hillman Cancer Center, University of Pittsburgh, Pittsburgh, PA 15232, USA; 2Epidemiology and Biostatistics Concentration, College of Public Health, University of South Florida, Tampa, FL 33620, USA; 3Department of Medical Microbiology and Parasitology, College of Health Sciences, Bayero University, Kano 700006, Nigeria; 4Department of Clinical Sciences and Community Health, University of Milan, 20133 Milan, Italy; 5Center for Health Data Science and Analytics, Houston Methodist Research Institute, Houston, TX 77030, USA; 6Faculty of Clinical Sciences, College of Health Sciences, Bayero University, Kano 700006, Nigeria; 7Institute for Molecular Bioscience, School of Biomedical Sciences, University of Queensland, Brisbane, QLD 4072, Australia; 8Department of Epidemiology, UCLA Fielding School of Public Health and Jonsson Comprehensive Cancer Center, Los Angeles, CA 90095, USA; 9Cancer Risk Factors and Life-Style Epidemiology Unit, Institute for Cancer Research, Prevention and Clinical Network, ISPRO, 50139 Florence, Italy; 10Department of Gastroenterology and Pancreato-Hepatobiliary, Tam Anh General Hospital, Hanoi 100000, Vietnam; 11Medical Informatics Center, Peking University, Beijing 100871, China; 12Department of Clinical Epidemiology, N.N. Blokhin National Medical Research Center for Oncology, 115478 Moscow, Russia; 13Department of Epidemiology, Harbin Medical University, Harbin 150081, China; 14Department of Epidemiology and Environmental Health, School of Public Health and Health Professions, University at Buffalo, Buffalo, NY 14214, USA; 15Section of Hygiene, University Department of Life Sciences and Public Health, Università Cattolica del Sacro Cuore, 00168 Roma, Italy; 16Department of Woman and Child Health and Public Health - Public Health Area, Fondazione Policlinico Universitario A. Gemelli IRCCS, 00168 Roma, Italy; 17Epidemiology and Prevention Group, Center for Public Health Sciences, National Cancer Center, Tokyo 104-0045, Japan; 18National Institutes of Biomedical Innovation, Health and Nutrition, Tokyo 162-8636, Japan; 19Department of Medicine, Memorial Sloan Kettering Cancer Centre, New York, NY 10065, USA; 20Department of Public and Community Health, School of Public Health, University of West Attica, 115 21 Athens, Greece; 21Department of Hygiene, Epidemiology and Medical Statistics, School of Medicine, National and Kapodistrian University of Athens, 115 27 Athens, Greece; 22Department of Epidemiology, Harvard T.H. Chan School of Public Health, Boston, MA 02115-5810, USA; 23Division of Cancer Epidemiology and Genetics, National Cancer Institute, National Institutes of Health, Rockville, MD 20892, USA; 24Centro Internacional de Pesquisa, A. C. Camargo Cancer Center, 01509-010 São Paulo, Brazil; 25EPIUnit, Instituto de Saúde Pública da Universidade do Porto, 4050-600 Porto, Portugal; 26Laboratório para a Investigação Integrativa e Translacional em Saúde Populacional (ITR), 4050-600 Porto, Portugal; 27Departamento de Ciências da Saúde Pública e Forenses e Educação Médica, Faculdade de Medicina, Universidade do Porto, 4200-319 Porto, Portugal; 28Department, Instituto de Investigación Sanitaria y Biomédica de Alicante, Universidad Miguel Hernandez (ISABIAL-UMH), 46020 Alicante, Spain; 29Consortium for Biomedical Research in Epidemiology and Public Health (CIBERESP), 28029 Madrid, Spain; 30Stony Brook Cancer Center, Stony Brooke University, Stony Brook, NY 11794, USA; 31Department of Medical and Surgical Sciences, University of Bologna, 40138 Bologna, Italy; 32Department of Epidemiology, School of Public Health, University of Pittsburgh, Pittsburgh, PA 15261, USA

**Keywords:** Gastric ulcers (Gus), duodenal ulcers (DUs), risk factors, gastric cancer

## Abstract

**Simple Summary:**

Gastric cancer (GC) is the fifth most common type of cancer and the fourth most common cause of cancer-related mortality. In this meta-analysis, we utilized SToP consortium data to investigate the association between gastric ulcer (GU) and duodenal ulcer (DU) and development of GC. Among 4106 GC cases and 6922 controls, we detected a positive association between GU and GC (OR = 3.04, 95% CI: 2.07–4.49). On the other hand, no significant association between DU and GC was detected (OR = 1.03, 95% CI: 0.77–1.39). In the pooled analysis, incorporating 11 case–control studies revealed positive association between the gastric ulcer and risk of gastric cancer.

**Abstract:**

Background. Gastric cancer (GC) is the fifth most common type of cancer and the fourth most common cause of cancer-related mortality. Although the risk of GC and peptic ulcer disease (PUD) is known to be increased by *H. pylori* infection, evidence regarding the direct relationship between PUD and GC across ethnicities is inconclusive. Therefore, we investigated the association between PUD and GC in the Stomach cancer Pooling (StoP) consortium. Methods. History of peptic ulcer disease was collected using a structured questionnaire in 11 studies in the StoP consortium, including 4106 GC cases and 6922 controls. The two-stage individual-participant data meta-analysis approach was adopted to generate a priori. Unconditional logistic regression and Firth’s penalized maximum likelihood estimator were used to calculate study-specific odds ratios (ORs) and 95% confidence intervals (CIs) for the association between gastric ulcer (GU)/duodenal ulcer (DU) and risk of GC. Results. History of GU and DU was thoroughly reported and used in association analysis, respectively, by 487 cases (12.5%) and 276 controls (4.1%), and 253 cases (7.8%) and 318 controls (6.0%). We found that GU was associated with an increased risk of GC (OR = 3.04, 95% CI: 2.07–4.49). No association between DU and GC risk was observed (OR = 1.03, 95% CI: 0.77–1.39). Conclusions. In the pooled analysis of 11 case–control studies in a large consortium (i.e., the Stomach cancer Pooling (StoP) consortium), we found a positive association between GU and risk of GC and no association between DU and GC risk.

## 1. Introduction

Gastric cancer (GC) is the fifth most common type of cancer and the fourth most common cause of cancer-related mortality, with more than 730,000 deaths reported worldwide [1,2]. Chronic inflammation with *Helicobacter pylori* (*H. pylori*) is implicated in the gastric carcinogenesis [3]. The risk of GC is associated with the extent of inflammation and severity of gastritis [4]. 

While both GC and peptic ulcer disease (PUD) are known to be associated with *H. pylori* infection, their definite causal mechanisms differ. GC results from uncontrolled proliferation of the epithelial cells and is accompanied by low-acid secretion (or hypochlorhydria) [5]. On other hand, gastric ulcer (GU) and duodenal ulcer (DU) are caused by the disruption of normal wound-healing processes in the gastric epithelial layer and are associated with hyperchlorhydria [6]. 

Although the risk of both GC and PUD is known to be increased by *H. pylori* infection, evidence regarding the direct relationship between PUD and GC is inconclusive. For example, in a cohort study of 57,936 patients from the Swedish Inpatient Register, Hansson et al. [7] found that among patients hospitalized for GU the risk of GC was almost twice as expected (standardized incidence ratio, SIR = 2.0, 95% confidence interval, CI: 1.8–2.4). On the contrary, they did not observe an association between DUs and the risk of GC (SIR = 0.9, 95% CI: 0.7–1.1). Results from two studies in Japan [8,9] showed a similar pattern with a positive association between GU and GC but no association between DU and risk of GC. However, the association between GU and GC was not ubiquitous across studies. In a large cohort study involving 93,229 US veterans (i.e., 4147 GC patients and 89,082 controls), Molloy et al. [10] found no association between GU and GC (odds ratio, OR = 1.02, 95% CI: 0.67–1.56).

While different studies had reported DU to be inversely associated with the risk of gastric tumorigenesis (e.g., SIR=0.6, 95% CI: 0.4–0.7 by Hansson et al. [7] and OR = 0.68, 95% CI: 0.47–0.95 by Molloy et al. [10]), a more recent study by Cho et al. [11] in South Korea found a positive association between DUs and risk of GC. This finding appears to be supported by the fact that in Asia where GC is more common, its rate of co-existence with DUs also tends to be high (e.g., found in 2–7% in China, Korea, and Taiwan) [11,12,13,14,15]. 

Given the excessive burden of both GC and PUD in many parts of the world and the inconsistency of results from existing individual studies regarding their relationship, it is imperative to conduct a study using large data from diverse populations to elucidate the true nature of their associations. The Stomach cancer Pooling (StoP) Project [16], a consortium of GC studies from across the world, provides a unique opportunity to perform such a study, through individual-level data that was harmonized to produce a more homogeneous definition of participants’ characteristics. In addition, the large dataset promises adequate power for valid subgroup analyses. Therefore, in the current study, we investigated the association between PUDs and the development of GC by pooling individual data from case–control studies in the StoP consortium. 

## 2. Methods

### 2.1. Study Population

The detailed methodology of the StoP Consortium has been previously published and its status has been recently updated [11,12]. Briefly, data for the current analysis were based on the second release of the StoP Project consortium database, which included 31 case–control and nested case–control studies on GC worldwide. A total of 11 participating studies with data availability on GU or DU were included in our analysis. Three of the studies were from Italy [13,14,15], one from Greece [17], one study from Russia [16], four from China [18,19,20,21], and Japan [22], and one study from the USA [23]. The study inclusion flowchart is summarized in Appendix A. The original data from each study were obtained after a signed data transfer agreement was given by the principal investigators. The consortium harmonized all data based on a predetermined format. The University of Milan Institutional Review Board (IRB) provided the ethical approval for the StoP project in April 2015.

Incident GC was defined by histologically confirmed diagnosis of GC. Seven [13,14,15] of the included studies also captured data on cancer anatomical subsite (i.e., cardia and non-cardia), and five [13,14,15] on histologic subtype (i.e., intestinal, diffuse, and others, including mixed, undifferentiated, and unclassified type). The main outcome for the current analysis was any type of GC, regardless of a subsite or histologic classification. We used cancer subsite and histological subtype each, as a polytomous outcome, to evaluate differences by subsite and GC histology, respectively, in the association between PUDs and GC. 

Controls were recruited from the same source of population and within the same enrollment periods as the cases, all without prior history of GC. In six studies [15,16,17,19,22,23], the controls were patients with various non-cancer diagnoses selected at the same health facility as the cases. The controls in the Japanese study were recruited from participants in a health checkup program [22]. Four studies [11,17,18,19] selected controls from the general population. In one study [18], 9% of eligible controls did not participate or were lost to follow up (See Appendix A). 

Data on the main exposures (i.e., history of PUDs diagnosed by a health professional or treated) were collected using a structured questionnaire in 11 studies. Studies without data on GU or DU were excluded from each respective analysis. The 10 studies included in the analysis with GU as the main exposure variable were Italy 1 [14], Italy 3 [15], Italy 4 [13], Russia [16], China 1 [19], China 2 [21], China 3 [18], China 4 [20], Japan 3 [22], and USA 1 [23]. On the other hand, 7 studies involved in the DU analysis were Italy 1 [14], Italy 4 [13], Greece [17], Russia [16], China 3 [18], China 4 [20], Japan 3 [22]. The questionnaires were administered by trained interviewers in all the studies except the Russian study [16] – for which it was self-administered – and two others [22,23] – for which the information on the procedure used to administer the questionnaire was not provided. Exposure data were collected in the same way and within the same period for cases and controls in all the studies.

### 2.2. Statistical Analysis and Covariates

The two-stage individual-participant data meta-analysis approach was adopted a priori. In the first stage, using unconditional logistic regression and Firth’s penalized maximum likelihood estimator [24], we calculated study-specific odds ratios (ORs) and 95% confidence intervals (CIs) separately for the association between each peptic ulcer type (i.e., GU and DU) and GC. Subject to availability, acceptable missing data (i.e., no more than 30%) and suitability of use (e.g., adequate data in categories of variables for models to converge), the logistic regression models were adjusted for sex, race/ethnicity (i.e., White, Black/African American, Asian, Hispanic/Latino, and other races), age (i.e., <55, 55–65, and >65 years), highest education completed (i.e., less than high school, high school and college/graduate), socioeconomic status (SES) (i.e., low, intermediate and high), body mass index (BMI) (i.e., < 18.5, 18.5-<23, 23-<25, 25-<27.5, 27.5–30, and >30 kg/m^2^), tobacco smoking status (i.e., never, former, and current (low: ≤10 cigarettes/day; intermediate/high: >10 cigarettes/day), level of alcohol consumption (i.e., never, low: ≤12 grams/day, moderate/high: >12 grams per day), *H. pylori* infection serostatus, fruit/vegetable intake (low, intermediate, and high, based on study-specific tertiles), history of the other PUD type (e.g., DU for the GU analysis), gastritis, other gastric diseases, and study site (for studies with multiple sites). Some variables were re-defined to avoid scant data in some studies and in subgroup analyses. Specifically, the intermediate/high smoking category of the smoking variable was formed by merging the corresponding categories in the original variable. The latter were further combined with the “low” category to form the 3-level smoking status variable used in the stratification analyses. BMI and SES were used as dichotomous in subgroup analyses. 

We employed multiple imputation by fully conditional specification [25] to replace the infrequent missing data in study covariates. Twenty imputed datasets were generated for each study under the assumption of missing at random, substituting the missing observations with values drawn from separate conditional distribution of each imputed variable. A logistic regression model was fitted on each of the 20 imputed datasets to obtain estimates, which were combined using Rubin’s rule [26] to produce study-specific regression coefficients and standard errors. 

The imputation and analysis models contained the study outcome and same set of covariates. We checked the robustness of the imputation by examining differences between the imputed and the original datasets, compared both their distributions using Kernel density plots, and the main analyses results produced by them [25]. The study-specific regression coefficients were then pooled in the second stage of the meta-analysis, using inverse-variance weighted random effect models to produce summary ORs and 95% CI risk estimates.

We conducted stratified analyses to assess potential difference in the association between each PUD and GC across levels of several participants’ characteristics (chosen not based on hypothesis), including age, sex, education, SES (low/intermediate vs. high), BMI (i.e., <25 vs. ≥25 kg/m^2^), smoking status, level of alcohol consumption, fruit/vegetable intake, *H. pylori* infection serostatus, history of type 2 diabetes, source of study controls (i.e., hospital versus general population), and geographical region. Statistical significance of differences in pooled OR estimates across strata was assessed in meta-regression models. Heterogeneity between studies was examined using the Method-of-Moments estimator and quantified by *I*^2^ (proportion of total model variance due to between-study variability) [27].

We also performed analyses with cancer subsite and cancer histological subtype as outcomes, to evaluate differences in the PUD-GC association across levels of these characteristics. Here, we fitted separate polytomous logistic regression models to estimate the outcome specific ORs for each study in the analysis phase of the first stage, following multiple imputation. The statistical significance of the differences in the pooled odds of an ulcer across levels of each polytomous outcome (i.e., cancer subsite and histology) was also tested in meta-regression models. 

### 2.3. Sensitivity Analysis 

We performed sensitivity analyses to test the consistency of our results. First, we adopted the one-stage individual participant data meta-analysis approach, where we fitted mixed-effect logistic regression models using data from all relevant studies, with random intercept and slope for the study indicator. These models included only covariates available in all the studies and missing data were replaced using the same approach as the main analysis. Second, we tested the influence of individual studies by omitting one study at a time in the pooling stage of the analysis. Third, to rule out the potential effects of differences in exposure data collection on our analysis, we excluded the studies that used self-administered questionnaire to collect data on the history of ulcers [15], and two others [20,21] for which we could not confirm if professional interviewers were used to administer the questionnaire. In addition, we restricted the analyses to studies with data on *H. pylori* infection serostatus [15,19,20] and compared results with and without it, to evaluate the effect of non-inclusion of the variable in the logistic regression models. Moreover, on the assumption that *H. pylori* infection is a necessary cause of GC, we fitted logistic regression models with *H. pylori* seropositive controls only. Finally, we compared DU and GU analyses using same set of studies (i.e., those with data on all the ulcer types) [11,12,15,16,18,20], to ensure the results we observed for the two ulcers were not affected by non-overlap of data. First stage statistical analysis was performed using the SAS (version 9.4)’s LOGISTIC, MI, and MI ANALYZE procedures, with a macro specifically developed for this purpose. The META and METAREG packages in Stata (version 16) were used to fit the meta-analysis models in the second (pooling) stage. All statistical tests were considered significant at the 0.05 level.

## 3. Results

Overall, the StoP Consortium included 4106 GC cases and 6922 controls with available data on either GU or DU. GC cases were older at the time of enrollment and were more likely to be male. In addition, after removing subjects with missing data, GC patients had higher rate of moderate/high alcohol drinking (59.6%) compared to controls (52.6%). Similarly, the rate of ever-smoking history was higher among GC subgroup (55.5% vs. 51.3%). The GC cases were more likely to be from low SES in comparison with controls (60.4% vs. 49.0%) (Table 1). 

Of total enrolled individuals, data on history of GU was available for 3868 cases and 6662 controls, data on history of DU for 3221 cases and 5260 controls. In order to study the association between PUD and GC, 10 studies qualified for GU (including 5 Asian studies, 4 European studies, and one study from the US). Likewise, seven studies were considered to investigate the association between DU and GC (including 4 European, and 3 Asian studies). Of note, Italy 3 study also had available data on DU; however, because none of the controls, and only one GC case, had DU, this study was not considered in the pooled analysis for DU. The overview of the studies is presented in Appendix A.

The distribution of patients with GU or DU across GC and control subgroups are shown per each study on Appendix A.

Among subjects with available GU data, 487 GC cases (12.6%) and 276 controls (4.0%) had a history of GU **(Table 1)**. The summary OR of GC for history of GU was 3.04 (95% CI, 2.07–4.49) (Figure 1). In GC cases with available DU data, 252 patients (7.8%) had a positive history of ulcer whereas in controls with available DU data, 318 subjects (6.0%) had a positive history for DU. The summary OR for history of DU was 1.03 (95% CI, 0.77–1.39) (Figure 2).

In stratified analyses the positive association between GU and GC tended to be stronger among non-cardia than cardia GC cases (OR = 2.50, 95% CI: 1.94–3.23; and OR = 1.43; 95% CI: 0.85–2.40; *P_interaction_ = * 0.05). Results were consistent in the other subgroups analyzed (Table 2).

In stratified analyses of the association between DU and GC, history of DU was inversely related with GC risk in individuals with high fruit or vegetable intake (OR: 0.64, 95% CI: 0.41–0.99) but not in those with low or intermediate intake (*P_interaction_* = 0.03). No significant association between pre-existing DU and the risk of GC was detected across various subgroups (Table 3).

### Sensitivity Analysis

Results from the one-stage analyses were similar to those from the two-stage analyses (OR = 3.14, 95% CI: 2.16–4.82) for GU main analysis; OR = 1.05, 95% CI: 0.95–1.24 for DU main analysis). Likewise, no substantial change in magnitude or statistical significance of the results were observed due to the exclusion of any of the studies at a time - no results were higher or lower than 10 percent of the corresponding full results for both ulcers. 

Results were also similar for both ulcer main analyses while excluding studies that had certainly (or probably) not used professional interviewers to collect data on ulcer history, although association was slightly stronger for GU (OR = 3.64, 95% CI: 2.09–6.36) compared to the full analysis, but the difference was not statistically significant. The OR and 95% CI for DU was 0.99 (95% CI: 0.66–1.48). 

Risk of GC was similar when adjusting for *H. pylori* in studies with data on infection serostatus. Odds ratio and for GU was 3.17 (95% CI: 1.62–6.22) in the adjusted model vs. 3.23 (95% CI: 1.71–6.12) in the unadjusted, while DU ORs and 95% CIs were 1.16 (0.48–2.80) and 1.21 (0.61–2.53), respectively for the *H. pylori* adjusted and unadjusted models. The results were also similar for both GU and DU versus the corresponding main analyses when the logistic models were fitted with *H. pylori*-positive controls only. The OR was 3.03 (95% CI: 2.02–4.76) for GU and 0.99 (95% CI: 0.67–1.40) for DU. 

No difference was observed in the conclusions on overall association between GC and either ulcer type when the analyses were conducted on the same set of studies for both ulcers. However, while the results for DU (OR = 1.03, 95% CI: 0.74–1.41) were almost the same as in the overall analysis, the OR for GU (2.47, 95% CI: 1.69–3.62) was slightly lower than that from the main analysis. 

## 4. Discussion

In a pooled analysis of 4,106 GC cases and 6,922 controls in the StoP Consortium, we found a positive association between GU and risk of GC. This association was more profound amongst individuals with non-cardia tumors. On the other hand, the pooled data analysis did not show any significant association between DU and the risk of GC. 

The overall patterns of the associations were in line with some previous reports on gastric carcinogenesis among patients with GU and DU [28,29]. The population-based long term follow-up study by Hansson et al. reported approximately two-fold increased risk of GC in subjects with GU [7]. For instance, an Italian case–control study reported an increased risk of GC following GU but not after DU [28]. The positive association between GU and GC was only noted in non-cardia tumors. This may be related to the divergent tumorigenesis pathways in these two anatomical locations. Overall, most previous evidence highlighted the association between GU and risk of non-cardia GC rather than tumors in cardia [30]. Interestingly, similar to our findings this study reported a stronger association between GU and GC in younger individuals. This may partly be explained by the fact that some early GC cases are misdiagnosed or misclassified at early stages as benign lesions [31]. Study by Hosokawa et al. in Japan unraveled an approximate 25% of GC cases were misdiagnosed and this phenomenon was amplified in lesions located in lesser curvature or posterior wall of stomach [32,33]. 

The association between GU and GC is partly attributed to similarities in risk factors and precursor states [7,34]. There are a number of common risk factors involved in pathophysiology of both GU and GC namely, atrophic gastritis, *H. pylori* infection, lower SES and smoking [35]. In this pooled analysis, we found a three-fold increase in the risk of GC among individuals with history of GU and the association was stronger among those with lower SES. Overall, an association between neighborhood SES and incidence of all types of GC is recognized. This association is specifically noted in non-cardia GC [36]. The lower SES is closely associated with several risk factors involved in both GC and GU, including exposure to smoking or *H. pylori* infection. The strong association between GU and GC development among intermediate and low SES may be partly explained by the presence of other risk factors. In addition to commonality in risk factors, PUD is shown to mediate the carcinogenic impact of some of the established GC risk factors. Previous study by the SToP consortium revealed the mediation role for PUD and a number of risk factors in development of GC [37]. The PUD was responsible for mediating 36% of the effect of tobacco smoking on gastric carcinogenesis [37].

Overall, the positive association between GU and incidence of GC was detected in eight out of ten studies. Across the continents, we had a wide range for the significant ORs and 95% CIs from 1.92 (1.09–3.37) in Japan 3 study to 12.51 (5.05–31.01) in China 4 study. This variation may partially relate to each study policy in subjects’ enrollment as well as the race/ethnicity of the selected populations. Disparities in disease presentation and tumor location across the races/ethnicities have been reported to play in a role in the pace of progression from the precancerous lesions to the GC [38]. Factors such as infectious etiologies [39] and divergent tumor biology [40,41] may contribute to this variation. 

Regarding the DU and GC, our pooled analysis did not show any significant association. However, in the stratified analysis, in individuals with dietary pattern of high fruit and vegetable intake, the DU cases had around 36% reduced risk of GC. Dietary features such as fresh fruits and vegetable intake are shown to reduce the risk of GC [42]. The DU has a complicated pathophysiology that entails altered acid secretion due to *H. pylori* infection as well as disturbed duodenal bicarbonate secretion due to the mucosal damage in duodenum [43]. Vegetables are rich in fiber, folate, selenium and carotenoids while fresh fruits provide antioxidants and vitamin C [44]. These contents may explain the strong protective effect of high vegetable and fruit consumption on the development of GC in individuals with DU.

This pooled analysis had a number of limitations. Due to different data acquisition strategies, some of the variables were not consistently captured in all studies, including information of PUD. Another limitation was the large missing data (>60%) on *H. pylori* infection as a key underlying factor in gastric carcinogenesis. To calculate the pooled OR for GC in GU with or without *H. pylori* infection, data from only three studies were qualified for GU and two studies qualified for the DU. When stratified based on *H. Pylori* serostatus, the positive association between GU and GC did not show significant difference (*P_interaction_* = 0.19). The considerable amount of missing data may have affected these findings. On the other hand, this study had a number of strengths, including the diverse ethnic and geographical distribution of study participants which enabled us to study the association between PUD and risk of GC across races/ethnicities. 

## 5. Conclusion

In conclusion, in this pooled analysis in the StoP Consortium, we found a positive association between GU and risk of GC which was more pronounced in non-cardia GC. These findings may further stress the similarities in the pathophysiologic pathway of GC and GU. However, no significant association between DU and risk of GC was detected. These findings may provide further insight into risk-stratification of pre-malignant lesions and lead to more a more efficient screening for GC.

## Figures and Tables

**Figure 1 cancers-14-04905-f001:**
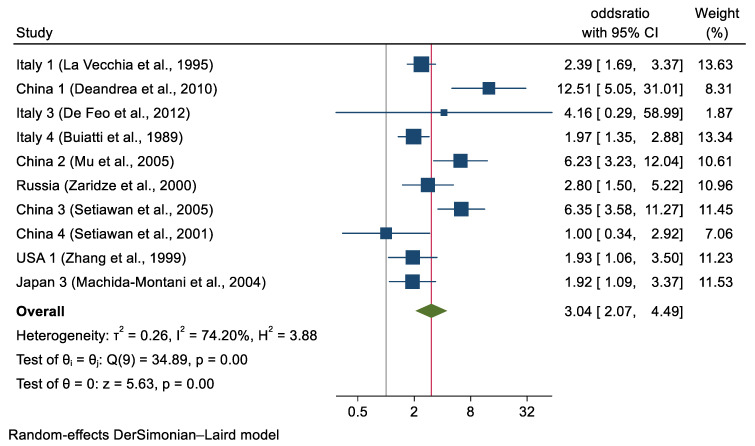
Study-specific and Pooled ORs for the Overall Association Between History of Gastric Ulcer and Gastric Cancer.

**Figure 2 cancers-14-04905-f002:**
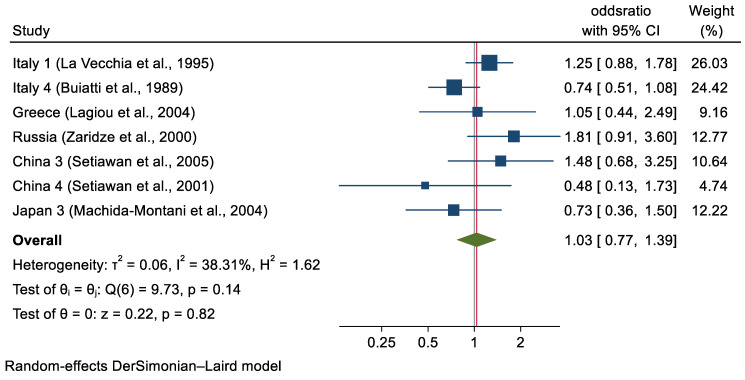
Study-specific and Pooled ORs for the Overall Association Between History of duodenal Ulcer and Gastric Cancer.

**Table 1 cancers-14-04905-t001:** Baseline Characteristics of Gastric Cancer Cases and Controls in the Stomach Cancer Pooling (StoP) Consortium.

	Gastric Cancer Cases (n = 4106)	Controls (n = 6922)	*p*-Value
Age (Mean ± SD)	61.5 (10.7)	57.0 (12.4)	<0.0001
Age categories (years)			<0.0001
≤55	1080 (26.3)	2881 (41.6)	
55–65	1206 (29.4)	1872 (27.0)	
≥65	1820 (44.3)	2169 (31.3)	
Sex			0.003
Male	2590 (63.1)	4171 (60.3)	
Female	1516 (37.0)	2751 (39.7)	
Race/ethnicity			0.59
White	545 (13.3)	683 (9.8)	
Black/African American	4 (0.1)	8 (0.1)	
Asian	8 (0.2)	5 (0.1)	
Hispanic/Latino	6 (0.2)	7 (0.1)	
Other	0 (0.0)	1 (0.01)	
Missing	3543 (86.3)	6218 (89.8)	
Education (completed)			<0.0001
Less than high school	1946 (47.4)	2147 (31.0)	
High school	812 (19.8)	1436 (20.8)	
College graduate	178 (4.3)	348 (5.0)	
Missing	1170 (28.5)	2991 (43.2)	
Socioeconomic status			<0.0001
Low	2418 (58.9)	3227 (46.6)	
Intermediate	1244 (30.3)	2360 (34.1)	
High	337 (8.2)	1004 (14.5)	
Missing	107 (2.6)	331 (4.8)	
Smoking			<0.0001
Never	1801 (43.9)	3345 (48.3)	
Former	871 (21.2)	1261 (18.2)	
Current, Low *	345 (8.4)	688 (10.0)	
Current, Intermediate/High *	1030 (25.1)	1575 (22.8)	
Missing	59 (1.4)	53 (0.8)	
Alcohol Drinking			
Never	769 (18.7)	1659 (24.0)	<0.0001
Low **	464 (11.3)	877 (12.7)	
Moderate/High **	1822 (44.4)	2817 (40.7)	
Missing	1051 (25.6)	1569 (22.7)	
Body Mass Index (Kg/m^2^)			
<18.5	239 (5.8)	191 (2.8)	<0.0001
18.5–23	1249 (30.4)	1917 (27.7)	
23–25	689 (16.8)	1272 (18.4)	
25–27.5	543 (13.2)	1137 (16.4)	
27.5–30	330 (8.0)	651 (9.4)	
>30	877 (21.4)	1148 (16.6)	
Missing	179 (4.4)	606 (8.8)	
Vegetables/fruits intake			<0.001
Low	1268 (30.9)	1845 (26.7)	
Intermediate	1319 (32.1)	2186 (31.6)	
High	1363 (33.2)	2379 (34.4)	
Missing	156 (3.8)	512 (7.4)	
History of diabetes			0.30
No	3180 (77.5)	5115 (73.9)	
Yes	210 (5.1)	307 (4.4)	
Missing	716 (17.4)	1500 (21.7)	
History of peptic ulcer			<0.0001
No	2,735 (66.6)	5122 (74.0)	
Yes	545 (13.3)	475 (6.9)	
Missing	826 (20.1)	1325 (19.1)	
History of GU			
No	3381 (82.3)	6386 (92.3)	<0.0001
Yes	487 (11.9)	276 (4.0)	
Missing	238 (5.8)	260 (3.8)	
History of DU			0.0003
No	3128 (76.2)	5377 (77.7)	
Yes	253 (6.2)	318 (4.6)	
Missing	725 (17.7)	1227 (17.7)	
*H. pylori* serostatus			<0.0001
Negative	393 (9.6)	612 (8.8)	
Positive	470 (11.5)	494 (7.1)	
Missing	3243 (79.0)	5816 (84.0)	

SD: standard deviation; BMI: body mass index; n: count; GU: gastric ulcer; DU: duodenal ulcer. * Low smoking: ≤10 cigarettes/day, Intermediate/High smoking: >10 cigarettes/day, ** Low drinking: ≤12 grams/day of alcohol consumption, Moderate/High drinking: >12 grams/day of alcohol consumption

**Table 2 cancers-14-04905-t002:** Association Between Gastric Ulcer and Gastric Cancer, Stratified by Selected Characteristics.

	Gastric Cancer				
	Cases	Controls			
	Gastric Ulcer (Yes)	Gastric Ulcer (Yes)	OR (95% CI)	*P_between-study_*	*P_interaction_*
Overall ^a^	487	276	3.04 (2.07–4.49)	0.00	N/A
Sex ^b^					0.37
Men	136 (27.9)	72 (27.6)	2.72 (1.82–4.07)	0.00	
Women	346 (72.1)	189 (72.4)	3.39 (1.98–5.63)	0.11	
Age (years) ^c^					0.15
≤55	137 (30.6)	84 (35.6)	4.22 (2.64–6.74)	0.08	
>55–<65	131 (29.2)	74 (31.3)	2.36 (1.66–3.35)	0.38	
≥65	180 (40.2)	78 (33.1)	2.89 (2.28–4.12)	0.00	
Education completed ^d^					0.97
Less than high school	139 (58.6)	51 (63.7)	3.53 (1.36–7.34)	0.03	
High school	76 (32.1)	23 (28.7)	3.64 (1.89–7.00)	0.24	
Graduate	22 (9.3)	6 (7.6)	3.16 (1.30–9.59)	0.74	
Socioeconomic status ^e^					0.34
Low/Intermediate	418 (87.4)	219 (79.9)	3.12 (2.03–4.78)	0.01	
High	60 (12.6)	55 (20.1)	2.24 (1.32–3.79)	0.96	
BMI (Kg/m^2^)					0.96
<25	291 (60.7)	165 (62.0)	2.99 (1.99–4.49)	0.01	
≥25	188 (39.3)	101 (38.0)	3.04 (1.99–4.67)	0.12	
Smoking status ^f^					0.14
Never	150 (34.1)	79 (35.7)	2.17 (2.07–3.46)	0.01	
Former	118 (26.8)	56 (25.4)	2.02 (1.41–2.90)	0.58	
Current	172 (39.1)	86 (38.9)	2.83 (2.29–4.21)	0.28	
Alcohol consumption ^g^					0.08
Never	72 (24.7)	51 (24.0)	4.53 (2.56–8.00)	0.05	
Low	65 (22.3)	43 (20.3)	2.71 (1.76–4.18)	0.56	
Moderate/high	154 (53.0)	118 (53.4)	2.07 (1.43–2.99)	0.14	
Fruit/vegetable intake ^h^					0.61
Low	129 (29.2)	67 (30.3)	3.36 (2.03–5.57)	0.05	
Intermediate	160 (36.2)	72 (32.6)	2.93 (1.97–4.37)	0.17	
High	153 (34.6)	82 (37.1)	2.50 (1.78–3.52)	0.67	
*H. pylori* serostatus ^i^					0.19
Positive	65 (52.0)	35 (55.6)	2.61 (1.38–4.95)	0.50	
Negative	60 (48.0)	28 (44.4)	4.58 (2.63–8.00)	0.24	
History of diabetes ^j^					0.98
Yes	12 (4.6)	12 (6.4)	2.88 (1.29–6.29)	0.42	
No	246 (95.4)	174 (93.6)	2.85 (1.94–4.28)	0.02	
Cancer subsite ^k^					0.05
Cardia	36 (16.6)	172 (50.0)	1.43 (0.85–2.40)	0.43	
Non-cardia	181 (83.4)	172 (50.0)	2.50 (1.94–3.23)	0.76	
Histological type ^l^					0.87
Intestinal	90 (47.9)	10 (33.3)	2.39 (1.53–3.73)	0.25	
Diffuse	40 (21.3)	104 (33.3)	2.04 (1.34–3.11)	0.88	
Other/unspecified	58 (30.8)	104 (33.3)	2.12 (1.45–3.09)	0.64	
Geographic region					0.30
Europe	207 (42.5)	90 (32.6)	2.28 (1.80–2.88)	0.74	
Asia	225 (46.2)	150 (54.3)	4.05 (1.87–8.74)	0.00	
Americas (USA only)	55 (11.3)	36 (13.0)	1.93 (1.06–3.50)	N/A	
Source of study Controls					0.84
Hospital	253 (51.9)	180 (65.2)	2.86 (1.82–4.50)	0.02	
Population	234 (48.1)	96 (34.8)	3.15 (1.43–6.94)	0.00	

^a^ Pooled ORs were obtained using random-effects models. ORs were adjusted, where possible (in a study), for age, sex, race/ethnicity, education, socioeconomic status, BMI, tobacco smoking status, level of alcohol consumption, fruit/vegetable intake, history of duodenal ulcer, history of diabetes, *H. pylori* serostatus and study site (for multicenter studies). ^b^ Variable was not usable for China 4 study due to sparse data in some categories of gastric ulcer and/or gastric cancer. ^c^ Variable was not available/unusable for Italy 3, China 3 and Japan 3 studies. ^d^ Studies considered: Italy 3, Italy 4, Russia, and China 3. ^e^ Variable not usable for the Italy 3 study due to too much missing data. ^f^ Variable not usable for the Italy 3 China 4 and Japan 3 studies due to sparse data in categories of exposure, outcome, or covariates. ^g^ Studies considered: Italy 1, China 1, Italy 3, Italy 4, USA 1 and Japan 3. ^h^ Variable not available for China 4, and unusable for Japan 3 study due to sparse data. ^I^ Studies considered: China 2, Russia, and Japan 3. ^j^ Studies considered: Italy 1, Italy 4, Russia, China 4, and Japan 3. ^k^ Studies considered: Italy 1, Russia, USA 1, and Japan 3. ^l^ Studies considered: Italy 3, Italy 4, Russia, and USA 1. Abbreviation: N/A: Not Applicabale; BMI: body mass index; CI: confidence interval; OR: odds ratio.

**Table 3 cancers-14-04905-t003:** Association between Duodenal Ulcer and Gastric Cancer, Stratified by Selected Characteristics.

	Gastric Cancer				
	Cases	Controls			
	Duodenal Ulcer (Yes)	Duodenal Ulcer (Yes)	OR (95% CI)	*P_between-study_*	*P_interaction_*
Overall ^a^	252	318	1.03 (0.77–1.39)	0.82	N/A
Sex					0.10
Men	186 (73.8)	237 (74.5)	0.91 (0.70–1.18)	0.37	
Women	66 (26.2)	81 (25.5)	1.43 (0.88–2.31)	0.26	
Age (years)					0.20
≤55	80 (31.7)	119 (37.4)	1.45 (0.96–2.21)	0.38	
>55–<65	74 (29.4)	89 (28.0)	1.05 (0.72–1.52)	0.87	
≥65	98 (38.9)	110 (34.6)	0.88 (0.62–1.25)	0.41	
Education completed ^b^					0.31
Less than high school	97 (61.0)	96 (58.5)	1.98 (0.71–5.51)	0.36	
High school	48 (30.2)	57 (34.8)	1.07 (0.63–1.81)	0.64	
Graduate	14 (8.8)	11 (6.7)	0.85 (0.56–1.29)	0.36	
SES ^c^					0.22
Low	136 (58.4)	173 (57.3)	0.94 (0.72–1.23)	0.56	
Intermediate	77 (33.0)	98 (32.4)	1.16 (0.71–1.87)	0.23	
High	20 (8.6)	31 (10.3)	1.77 (0.89–3.51)	0.89	
Socioeconomic status ^c^					0.25
Low/intermediate	213 (91.4)	271 (89.7)	1.70 (0.86–3.38)	0.11	
High	20 (8.6)	31 (10.3)	1.08 (0.78–1.52)	0.82	
BMI (kg/m^2^) ^d^					0.95
<25	141 (67.8)	183 (63.8)	0.99 (0.60–1.66)	0.02	
≥25	67 (32.2)	104 (36.2)	1.01 (0.70–1.47)	0.93	
Smoking status ^e^					0.52
Never	64 (27.9)	84 (30.3)	1.08 (0.89–1.84)	0.49	
Former	74 (32.3)	67 (24.2)	1.18 (0.76–1.81)	0.82	
Current	91 (39.8)	126 (45.5)	1.14 (0.63–1.40)	0.62	
Alcohol consumption ^f^					0.51
Never	33 (20.8)	50 (20.7)	1.02 (0.65–1.61)	0.24	
Low	19 (11.9)	25 (10.4)	0.85 (0.59–1.23)	0.26	
Moderate/high	107 (67.3)	166 (68.9)	1.41 (0.62–3.20)	0.31	
Fruit/vegetable intake ^g^					0.03
Low	81 (41.8)	74 (30.3)	1.61 (0.87–2.97)	0.11	
Intermediate	42 (21.7)	93 (38.1)	1.19 (0.75–1.88)	0.33	
High	71 (36.5)	77 (31.5)	0.64 (0.41–0.99)	0.39	
*H. pylori* serostatus ^h^					0.86
Positive	34 (75.6)	29 (69.0)	1.15 (0.48–2.74)	0.13	
Negative	11 (24.4)	13 (31.0)	1.29 (0.46–3.63)	0.65	
History of diabetes ^i^					0.89
Yes	11 (4.6)	15 (5.2)	1.03 (0.42–2.57)	0.77	
No	227 (95.4)	27 (94.8)	1.11 (0.80–1.53)	0.11	
Cancer subsite ^j^					0.10
Cardia	20 (10.5)	258 (50.0)	1.89 (1.03–3.47)	0.84	
Non-cardia	171 (89.5)	258 (50.0)	0.10 (0.63–1.57)	0.03	
Histological type ^k^					0.83
Intestinal	57 (48.7)	120 (33.3)	0.84 (0.55–1.29)	0.00	
Diffuse	20 (17.1)	120 (33.3)	0.88 (0.28–2.71)	0.45	
Other/unspecified	40 (34.2)	120 (33.3)	1.27 (0.36-4.48)	0.70	
Region					0.56
Europe	58 (23.0)	74 (23.3)	1.10 (0.75–1.61)	0.09	
Asia	194 (77.0)	244 (76.7)	0.89 (0.49–1.62)	0.25	
Source of study Controls					0.26
Hospital	125 (49.6)	170 (53.4)	1.20 (0.89–1.62)	0.34	
Population	127 (50.4)	148 (46.6)	0.85 (0.51–1.43)	0.21	

^a^ Pooled ORs were obtained using random-effects models. ORs were adjusted, where possible (in a study), for age, sex, race/ethnicity, education, socioeconomic status, BMI, tobacco smoking status, level of alcohol consumption, fruit/vegetable intake, history of diabetes, GU, gastritis and other gastric diseases, *H. pylori* serostatus and study site (for multicenter studies). **^b^** Variable was available/usable only for Italy 4, Russia, China 3, and China 4 studies. ^c^ Variable was not usable for the Greece study ^d^ Variable not usable for the China 3 study due to sparse data ^e^ Variable not usable for the Greece and China 4 studies due to sparse data. ^f^ Studies considered: Italy 1, Italy 4, and Japan 3. ^g^ Studies considered: Italy 1, Italy 4, Greece, and Japan 3. ^h^ Variable available/usable only for Russia and Japan 3 studies. ^I^ Variable not available for the China 3 study. ^j^ Studies considered: Italy 1, Italy 4, Russia, and Japan 3. ^k^ Studies with available/usable data: Italy 4, Russia. Abbreviation: BMI: body mass index; CI: confidence interval; OR: odds ratio.

## Data Availability

The data that support the findings of this study are available on request from the corresponding author. The data are not publicly available due to privacy or ethical restrictions.

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
