# Peer review of "The Association between Peptic Ulcer Disease and Gastric Cancer: Results from the Stomach Cancer Pooling (StoP) Project Consortium"

_cancers, 2022, doi:10.3390/cancers14194905_

Round 1

Reviewer 1 Report

This study aims to investigate the association between peptic ulcer disease and the development of Gastric Cancer using a structured questionnaire in several studies of the StoP consortium.

I have just a couple of suggestions:

In the statistical methods section, the sentences in lines 185-190 could be more detailed in describing the multiple imputations of the missing data method.

  For Tables 2 and 3, the authors might consider simplifying the data for cases and controls by entering a single percentage value rather than two.

 Overall, the article makes an important contribution to this field and is easy to read.

Author Response

Comment 1

In the statistical methods section, the sentences in lines 185-190 could be more detailed in describing the multiple imputations of the missing data method.

Response 1

We thank the reviewer for this comment. We have added a sentence (now lines 200-203) to make it clearer the multiple imputation of the missing data. 

Comment 2

For Tables 2 and 3, the authors might consider simplifying the data for cases and controls by entering a single percentage value rather than two.

Response 2

We have simplified the percentages in Tables 2 and 3.

Reviewer 2 Report

Dear authors,

Great congratulations on this submission. I carefully checked the content and it was perfect for publication in this journal. However, the following amendments should be followed before publishing this record.

1-   Please add a flowchart for inclusion/exclusion criteria associated with your data.

2-   Please develop your discussion using the updated references.

3-   If funnel plots are available, please supplement the relevant records.

4-   The respected authors mentioned that “two-stage individual-participant data meta-analysis approach…” has been used for this study. Would you please kindly explain the differences of this method in comparison to commonly used meta-analysis procedures?

5-   Please re-write the conclusion section and highlight the novel aspect of this study. The mentioned finding in this section was previously described in the results section. 

Author Response

Comment 1

Please add a flowchart for inclusion/exclusion criteria associated with your data.

Response 1

We have added a flowchart, following your comment.

Comment 2

Please develop your discussion using the updated references.

Response 2

The references were revisited, and a number of updated references were added according to this comment.

Comment 3

If funnel plots are available, please supplement the relevant records.

Response 3

Unfortunately, the funnel plots were not generated in the current analysis. 

Comment 4

The respected authors mentioned that “two-stage individual-participant data meta-analysis approach…” has been used for this study. Would you please kindly explain the differences of this method in comparison to commonly used meta-analysis procedures?

Response 4

The two methods are technically similar as they both have first and second stages. In fact, the second (meta-analytic) stages are implemented using the same procedure in most analysis software. The key difference between them is in the first stage, where the effect measures are generated. In the two-stage pooling, one has access to the actual individual data from all the studies, as such can harmonize variables and define them in the same way for each study, thereby eliminating heterogeneity in effect measure estimates that could arise due to between-study differences in analysis variables definition. In traditional meta-analysis, the investigator could only use the estimate(s) from original studies; has no control over the generation of the effect measures for the individual studies included in the meta-analysis, hence any errors or variations within them.

Comment 5

Please re-write the conclusion section and highlight the novel aspect of this study. The mentioned finding in this section was previously described in the results section.

Response 5

We thank the reviewer for this comment and have revised the conclusion accordingly.